# SARS-CoV-2 Seroprevalence Survey in Grocery Store Workers—Minnesota, 2020–2021

**DOI:** 10.3390/ijerph19063501

**Published:** 2022-03-16

**Authors:** Madhura S. Vachon, Ryan T. Demmer, Stephanie Yendell, Kathryn J. Draeger, Timothy J. Beebe, Craig W. Hedberg

**Affiliations:** 1Division of Environmental Health Sciences, University of Minnesota, Minneapolis, MN 55455, USA; hedbe005@umn.edu; 2Division of Epidemiology and Community Health, University of Minnesota, Minneapolis, MN 55455, USA; demm0009@umn.edu; 3Minnesota Department of Health, St. Paul, MN 55164, USA; stephanie.yendell@state.mn.us; 4Department of Agronomy and Plant Genetics, University of Minnesota, Minneapolis, MN 55455, USA; draeg001@umn.edu; 5Division of Health Policy and Management, University of Minnesota, Minneapolis, MN 55455, USA; beebe026@umn.edu

**Keywords:** COVID-19, grocery workers, serosurvey

## Abstract

Grocery workers were essential to the workforce and exempt from lockdown requirements as per Minnesota Executive Order 20–20. The risk of COVID-19 transmission in grocery settings is not well documented. This study aimed to determine which factors influenced seropositivity among grocery workers. We conducted a cross-sectional study of Minnesota grocery workers aged 18 and older using a convenience sample. Participants were recruited using a flyer disseminated electronically via e-mail, social media, and newspaper advertising. Participants were directed to an electronic survey and were asked to self-collect capillary blood for IgG antibody testing. Data were analyzed using logistic regression and adjusted for urbanicity, which confounded the relationship between number of job responsibilities in a store and seropositivity. Of 861 Minnesota grocery workers surveyed, 706 (82%) were tested as part of this study, of which 56 (7.9%) tested positive for IgG antibodies. Participants aged 65–74 years had the highest percent positivity. Having multiple job responsibilities in a store was significantly associated with seropositivity in our adjusted model (OR: 1.14 95% CI: 1.01–1.27). Workplace factors influenced seropositivity among Minnesota grocery workers. Future research will examine other potential factors (e.g., in-store preventive measures and access to PPE) that may contribute to increased seropositivity.

## 1. Introduction

In response to the COVID-19 pandemic, movements and interpersonal interactions were restricted to various degrees across the United States and around the world. However, essential workers, including workers in grocery stores, were exempted from quarantine measures and thus remained at higher risk for community transmission of COVID-19. In Minnesota, through November 2021, there have been 887,368 documented cases of COVID-19, 44,577 hospitalizations, and 9229 deaths. Confirmed daily case counts reached a peak in early November 2020 [1]. Seroprevalence in Minnesota during September 2020 was estimated to be 8.0% based on a nationwide seroprevalence survey using retained blood specimens in commercial laboratories [2].

Throughout the pandemic, grocery stores have remained one of the few indoor public places that people continued to visit while otherwise practicing social distancing. However, the risk of SARS-CoV-2 infection among grocery workers remains unknown.

Limited data are available on grocery store exposures or risk to grocery store workers. A 2020 study in a Massachusetts-based grocery store found that employees who worked in direct customer service jobs had a five-fold increase in the odds of PCR positivity for SARS-CoV-2 when compared to employees who had other job types; 76% of SARS-CoV-2 PCR positive cases were asymptomatic [3]. Compliance with public health prevention measures, including enforcing mask use in grocery stores and limiting the use and/or contact with reusable bags have been shown to be associated with decreased COVID-19 risk in grocery workers [4].

The purpose of this study was to establish prevalence estimates for antibodies to SARS-CoV-2 in Minnesota grocery store workers and determine what factors (workplace, household, or county-level) may have influenced seropositivity in this target population. Ensuring the safety of grocery workers is of high importance to both the food industry and the larger community. As workplaces operate in communities with varying levels of immunization, understanding the risk posed during continuous operation of grocery stores could be important to guide recommendations for managing community exposures in other settings. Additionally, the results of this study have store-level policy implications for worker safety, support, and compensation in the state of Minnesota.

## 2. Materials and Methods

We conducted a cross-sectional seroprevalence study of Minnesota grocery workers aged 18 and older by obtaining a convenience sample of grocery employees. This study was approved by the University of Minnesota Institutional Review Board and all participants provided informed consent. Because no comprehensive list of grocery store workers was available, participants were recruited using a flyer disseminated electronically via e-mail, shared on state health department social media, and included in newspaper advertising. We partnered with the United Food and Commercial Workers Union Local 1189 to distribute the flyer to union members statewide. We also collaborated with the University of Minnesota Regional Sustainable Development partnerships to mail the flyer to 250 grocery stores in communities with 2500 or fewer residents [5].

Participants were directed to an electronic survey on the flyer advertisement. Survey questions were developed by epidemiologists with extensive outbreak investigation experience (MSV, CWH) in consultation with survey design specialists (RTD, TJB) who conducted related studies [6,7]. Due to the time-sensitive nature of the survey, no formal validation of survey questions was conducted. The survey included informed consent and confirmation of study eligibility, questions about store characteristics, demographics, and physical health. Store characteristics included store name, store county, job responsibilities, hours worked per week, and length of time worked at the store location. We grouped job responsibilities into two categories: direct customer contact and no direct customer contact. Jobs classified as direct customer contact included front end coordinator, manager, assistant manager, customer service representative, cashier, bagger, curbside shopper/carry out, cart attendant, pharmacist, pharmacy technician, deli/meat/seafood specialist, produce worker, baker, florist, and barista. Demographic information included age group, gender identity, race, ethnicity, number of people in the household, and age groups in the household. Physical health information included pregnancy status, underlying health conditions, underlying health conditions in the household, history of COVID-19 symptoms, prior COVID-19 testing, and symptoms/testing in the household. We also requested the name and mailing address of study participants to send them sample collection kits.

Store county level factors of interest were county urbanicity, COVID-19 disease incidence in the county, and COVID test rate and positivity in the county. Definitions of urbanicity were taken from the Minnesota Center for Rural Policy and Development. Counties were categorized as “Entirely Rural”, “Town/Rural mix”, “Urban/Town/Rural mix”, and “Entirely Urban” [8]. COVID-19 incidence by county was taken from reports published by the Minnesota Department of Health [9].

Study participants were mailed a capillary blood self-collection kit (finger-stick) that included Neoteryx Mitra^®^ (Torrance, CA, USA) 10 µL samplers. Samples were tested using the Quansys Q-Plex™ (Logan, UT, USA) SARS-CoV-2 Human IgG (4-Plex). This qualitative enzyme-linked immunosorbent assay (ELISA) detects human immunoglobulin G (IgG) antibodies to both S1 and S2 spike proteins present in the blood sample [10]. The assays were performed at Quansys laboratories, (Logan, UT, USA) using microplate arrays with 4 spots in each well: (1) Recombinant SARS-CoV-2 Spike Glycoprotein (S1), (2) Recombinant SARS-CoV-2 Spike Glycoprotein (S2), (3) Sheep Fc, a negative control to ensure no cross-reactivity occurs between human IgGs in the sample and the Fc-Tag on the SARS-CoV-2 Spike proteins, (4) Anti-Human IgG, a positive control to ensure that sample was added to the plate and that the procedure was properly followed. Qualitative positive cutoffs for S1 and S2 were generated by multiplying a specific correction factor to the ratio of low to high calibrator signal [8]. Quansys (Logan, UT, USA) assay validation study reports an estimated sensitivity of 97% and specificity of 100%. Samples were tested from 8 December 2020–24 March 2021 [10].

Descriptive statistics included frequency analysis (percentages) for categorical variables and mean and standard deviation or median and interquartile range for continuous variables. Univariate comparisons were determined by the use of a parametric two sample *t*-test or non-parametric Mann–Whitney–Wilcoxon rank sum test, when appropriate, for continuous variables. For binary variables, comparisons were made using the Chi-squared test or Fisher’s exact test for variables with at least one cell count less than five. A one-way ANOVA was performed to compare the mean number of job responsibilities by rurality classification. In our primary analysis, using logistic regression, we estimated the association between number of job responsibilities in the store and SARS-CoV-2 seropositivity adjusted for having a household member who tested positive for COVID-19, urbanicity, age group, and background county case rate. Regression coefficients were exponentiated to obtain odds ratios (ORs), and exact 95% confidence intervals (CIs) were calculated. All data were analyzed using RStudio^®^ (Boston, MS, USA) statistical software.

## 3. Results

We received survey responses from 861 people and received samples for testing from 706 (82%) (Figure 1). Of these, 56 (7.9%) were positive. The majority of study participants were female (61%) and White (95%). Of our study participants, 4 (0.57%) were pregnant, 141 (20%) identified having an underlying condition, and 171 (24%) stated that someone in their household had an underlying condition (Table 1).

Having a prior positive diagnostic test was strongly associated with seropositivity in our study sample as was having a household member who tested positive for COVID-19 (Table 1). Of all study participants who were seropositive, 15 (26.8%) reported being asymptomatic since March 2020. Percent seropositivity by month of self-reported onset of COVID-19 symptoms peaked in October (Figure 2). Documentation of COVID-19 symptoms was associated with seropositivity. Participants aged 65–74 years had the highest percent seropositivity (Table 1).

There were 90 different stores represented in our sample. Study participants worked a median of 40 h per week [IQR: 32–40]. Of tested study participants, 385 (54.5%) stated that they only had a single job responsibility within a store, while 320 (45.3%) had multiple job responsibilities (e.g., cashier, bagger, carry-out). The number of job responsibilities for a single person ranged from 1–11. There were 54 (out of 87) Minnesota counties represented in our sample (Figure 1). Of our study participants, 37 (5%) resided in counties that were classified as “Entirely Rural”, 76 (11%) lived in counties classified as “Town/Rural mix”, 65 (9%) as “Urban/Town/Rural mix”, and 528 (75%) as “Entirely Urban” (Table 1). People residing in more rural counties had a higher percent seropositivity than those in urban counties.

One-way ANOVA indicated that there was a statistically significant difference in mean number of job responsibilities between at least two groups (F = 15.11, *p* < 0.0001). Tukey HSD tests for multiple comparisons found that mean number of job responsibilities was significantly higher in counties classified as “Town/Rural mix” when compared to “Entirely Rural” counties (*p* = 0.003). Mean number of job responsibilities was significantly lower in counties classified as “Urban/Town/Rural mix” when compared to “Town/Rural mix” counties (*p* < 0.0001). Mean number of job responsibilities was also significantly lower in counties classified as “Entirely Urban” when compared to “Town/Rural mix” counties (*p* < 0.0001).

The average number of job responsibilities worked within a store was higher among seropositive participants (3.3) compared to seronegative participants (2.6) (*p* = 0.05) In our sample, 593 (84.0%) grocery employees worked direct contact jobs. The rate of seropositivity among person working a direct contact job (8.6%) was more than twice the rate of seropositivity among persons without direct contact (3.6%); however, this difference was not statistically significant (Table 1).

Adjusted logistic regression indicated that having a household member who tested positive for COVID-19 was strongly associated with SARS-CoV-2 seropositivity (OR = 10.62, 95% CI: 5.35–21.05) (Table 2). Having more job responsibilities within grocery stores (OR = 1.14, 95% CI: 1.01–1.27) and the county case rate (OR = 1.24, 95% CI: 1.00–1.56) were also significantly associated with seropositivity among sampled Minnesota grocery workers.

## 4. Discussion

The results of our study highlight several potential COVID-19 transmission routes for Minnesota grocery store workers. Having multiple job responsibilities was associated with increased seropositivity, suggesting a risk for occupational exposure. This is supported by a rate of seropositivity that was twice as high among grocery store workers with direct customer contact than among workers without direct customer contact. Furthermore, COVID-19 case rates by county were associated with seropositivity among grocery store workers working in the same county. These findings suggest that performing multiple job tasks within the grocery store may increase the likelihood of sporadic respiratory exposure to SARS-CoV-2 virus in this occupational setting. The stronger relationship between seropositivity among grocery store workers and their household members is consistent with a higher risk of transmission with prolonged exposure in the household setting.

Rates of SARS-COV-2 seroprevalence in our sample were similar to concurrent population estimates [7]. Persons who were seropositive reported a history of illness onset that aligned with the fall surge in COVID-19 transmission in Minnesota [1]. Although the rate of seropositivity among study participants who worked direct customer service jobs was twice that of workers who did not have direct customer service, our findings were weaker than those reported in previous studies [3]. Unfortunately, our cross-sectional study design did not allow us to examine patterns of transmission within households or potentially between workers in a grocery store.

Several factors not directly addressed in this study may contribute to increased seropositivity among grocery workers. We were not able to evaluate the use of preventive resources, including plexiglass shields that function as a barrier between cashiers and customers, masks, soap, hand sanitizer, and self-pay options in grocery store locations. We were also unable to collect information about whether our study participants encountered any difficulties in complying with public health recommendations intended to prevent COVID-19 transmission. We have commenced a second round serosurvey in this same study sample that address questions about COVID-19 prevention measures, to better address approaches to prevent transmission to workers in similar retail settings.

## 5. Conclusions

Having multiple different job responsibilities may increase the risk of occupational exposure to SARS-CoV-2 among Minnesota grocery workers. There is a strong risk for transmission within grocery worker households, although whether that transmission occurs primarily from grocery store workers to household members, or vice versa cannot be determined from the present study.

## Figures and Tables

**Figure 1 ijerph-19-03501-f001:**
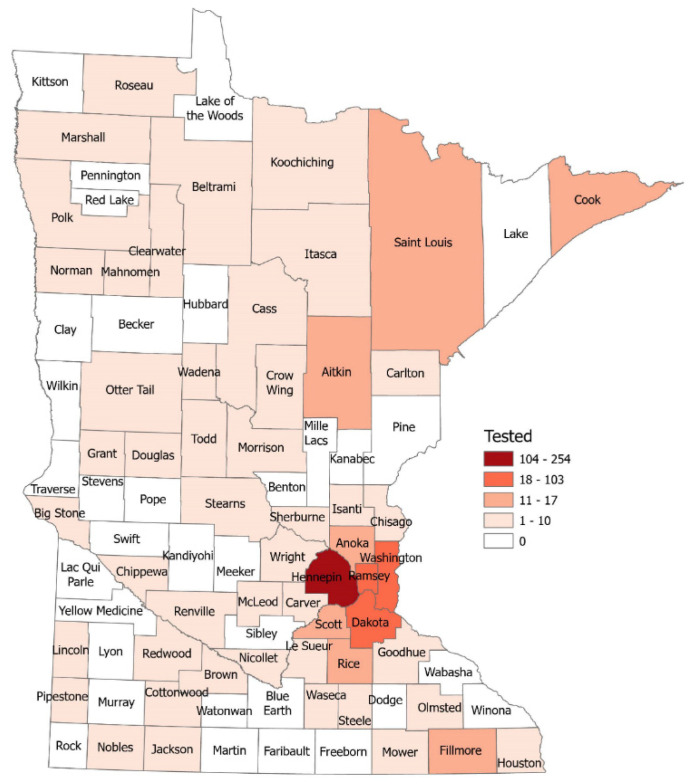
Map of tested study participants by county.

**Figure 2 ijerph-19-03501-f002:**
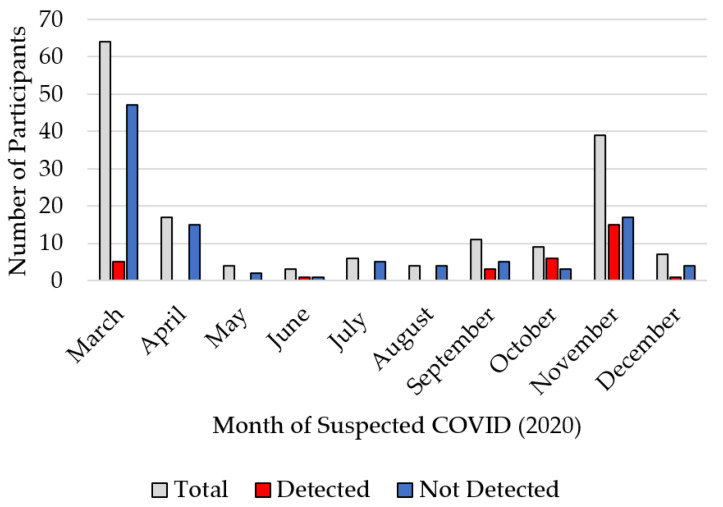
Number of study participants by month of suspected COVID-19 via self-reporting and serosurvey test results.

**Table 1 ijerph-19-03501-t001:** Demographic characteristics of study participants by SARS-CoV-2 seropositivity.

	Overall (*n* = 706)	Seropositive (*n* = 56)	% Seropositive	Estimated Odds Ratio	Lower 95% Confidence Limit	Upper 95% Confidence Limit	*p*
Age Group, *n* (%)							
18–24 years	66 (9.3)	7 (12.5)	10.6	(ref) ^1^			
25–34 years	169 (23.9)	11 (19.6)	6.5	0.59	0.22	1.66	0.29
35–44 years	149 (21.1)	5 (8.9)	3.4	0.29	0.08	0.95	0.04
45–54 years	142 (20.1)	12 (21.4)	8.5	0.78	0.30	2.18	0.62
55–64 years	150 (21.2)	17 (30.3)	11.3	1.07	0.44	2.91	0.88
65+ years	30 (4.2)	4 (7.1)	13.3	1.30	0.32	4.69	0.70
Gender Identity, *n* (%)							
Male	258 (36.5)	20 (35.7)	7.8	(ref) ^1^			
Female	431 (61.0)	36 (64.3)	8.4	1.08	0.62	1.95	0.78
Other Gender Identity (Transgender, gender variant/nonconforming, other)	17(2.4)	0(0.0)	0.0	7.6 × 10^−7^	NA	2.24 × 10^9^	0.98
Race, *n* (%)							
White or Caucasian	665 (94.2)	52 (92.9)	7.8	(ref) ^1^			
Black or African American	6 (0.9)	0 (0.0)	0	7.5 × 10^−7^	NA	4.51 × 10^34^	0.99
American Indian or Alaska Native	2 (0.3)	0 (0.0)	0	7.5 × 10^−7^	NA	3.52 × 10^109^	0.99
Asian	16 (2.3)	3 (5.4)	18.8	2.72	0.61	8.77	0.13
Native Hawaiian or Pacific Islander	1 (0.1)	0 (0.0)	1	7.5 × 10^−7^	NA	1.79 × 10^206^	0.99
Other	14 (2.0)	1 (1.8)	7.1	9.07× 10^−1^	0.05	4.69	0.93
Unknown	2 (0.3)	0 (0.0)	0	--			--
Ethnicity, *n* (%)							
Not Hispanic or Latino	651 (92.2)	53 (94.6)	8.1	(ref) ^2^			
Hispanic or Latino	17 (2.4)	1 (1.8)	5.9	0.71	0.02	4.72	1.00
Unknown	38 (5.4)	2 (3.57)	5.3				
County Urbanicity, *n* (%)							
Entirely Rural	37 (5.2)	5 (8.9)	13.5	(ref) ^1^			
Rural/Town Mix	76 (10.8)	8 (14.3)	10.5	0.75	0.23	2.66	0.64
Urban/Town/Rural Mix	65 (9.2)	7 (12.5)	10.8	0.77	0.23	2.79	0.68
Entirely Urban	528 (74.8)	36 (64.3)	6.8	0.47	0.19	1.43	0.14
Report of COVID-19 Symptoms since March 2020, *n* (%)							
No Symptoms	336 (47.6)	15 (26.8)	4.5	(ref) ^1^			
Symptoms	370 (52.4)	41 (73.2)	11.1	2.67	1.48	5.06	0.002
Household Member with a Prior Positive Diagnostic Test, *n* (%)							
No Positive Test	646 (91.5)	36 (64.3)	5.6	(ref) ^1^			
Positive Test	56 (7.9)	20 (35.7)	35.7	9.41	4.91	17.83	<0.0001
Unknown	4 (0.6)	0 (0.0)	0.0	--	--	--	
Pregnant, *n* (%)							
Not Pregnant	701 (99.3)	56 (100.0)	8.0				
Pregnant	4 (0.57)	0 (0.0)	0.0				
Unknown	1 (0.14)	0 (0.0)	0.0				
Underlying Condition, *n* (%)							
No	551(78.0)	41 (73.2)	7.4	(ref) ^1^			
Yes	141 (20.0)	14 (25.0)	9.9	1.37	0.70	2.53	0.33
Prefer not to Answer	13 (1.8)	1 (1.8)	7.7	--			
Unknown	1 (0.1)	0 (0.0)	0	--			
Job Type in Store, *n* (%)							
No Direct Contact	110 (15.6)	4 (7.1)	3.6	(ref) ^1^			
Direct Contact	593 (84.0)	51 (91.1)	8.6	2.49	0.99	8.38	0.08
Unknown	3 (0.42)	1 (1.8)	33.3	--			
Prior diagnostic test, *n* (%)	(*n* = 168)	(*n* = 16)					
Negative	147 (87.5)	8 (50.0)	5.4	(ref) ^2^			
Positive	11 (6.5)	7 (43.8)	63.6	28.7			<0.0001
Inconclusive	2 (1.2)	0 (0.0)	0.0	--			
Unknown, results pending	7 (4.2)	0 (0.0)	0.0	--			
Prefer not to answer	1 (0.6)	1 (6.3)	100.0	--			
Number of People in the Household, (mean, SD)	2.7(1.3)	2.8 (1.3)		1.03 ^1^	0.83	1.27	0.77
Number of Job Responsibilities in Store, (mean, SD)	2.7 (2.4)	3.3 (2.7)		1.12 ^1^	1.00	1.24	0.03
Hours Worked per Week, (median, IQR)	40.0 (8.0)	40.0 (15.0)		1.00 ^1^	0.98	1.03	0.90

^1^ Odds ratio estimates obtained from univariate logistic regression; first category is used as a reference category for categorical variables. ^2^ Odds ratio obtained from Fisher’s exact test.

**Table 2 ijerph-19-03501-t002:** Logistic regression analysis of the number of job responsibilities and seropositivity adjusted for urbanicity, age group, and county case rate.

Antibodies Detected	OR	95% CI	*p*-Value
Number of Job Responsibilities	1.14	(1.01, 1.27)	0.03
Household Member who Tested Positive for COVID-19	10.62	(5.35, 21.05)	<0.0001
Urbanicity (Reference: Entirely Rural)			
Town/Rural Mix	0.22	(0.05, 1.01)	0.05
Urban/Town/Rural Mix	0.42	(0.10, 1.75)	0.21
Entirely Urban	0.30	(0.10, 1.05)	0.04
Age Group (Reference: 18–24 years)			
25–34 years	0.70	(0.24, 2.16)	0.52
35–44 years	0.31	(0.08, 1.10)	0.07
45–54 years	0.79	(0.28, 2.40)	0.67
55–64 years	1.20	(0.45, 3.50)	0.73
65+ years	1.51	(0.33, 6.11)	0.57
County Case Rate	1.24	(1.00, 1.56)	0.05

## Data Availability

The data presented in this study are available on request from the corresponding author. The data are not publicly available due to privacy restrictions included in consent statements.

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
