# Peer review of "SARS-CoV-2 Seroprevalence Survey in Grocery Store Workers—Minnesota, 2020–2021"

_ijerph, 2022, doi:10.3390/ijerph19063501_

Round 1

Reviewer 1 Report

By virtue of this simple cross sectional study, the Authors have tried to establish the workplace factors those influenced seropositivity among Minnesota grocery workers. However the authors are requested to address the following critical issues:

  1. I am not cear by the instroduction of the manuscript why this study has been undertaken? What is the purpose of conducting this study?
  2. What extactly the authors wanted to explore by knowing the factors those already influenced the sero positivity?
  3. Will these factors further affected the vaccination?
  4. The conceptual framework in the introduction is completely misising
  5. Specific Objectives of the study and the study purpose is missing.
  6. Why was the convenient sampling technology has been adopted as the survey instrument was administered online (electronic survey was used)
  7. Was the survey instrument was already validated and published?
  8. If the survey instrument is developed by the authors, what about the reliability and validity of the survey instrument developed? These metrics and this critical information is completely missing in the manuscript.
  9. This information pertaining to the survey instrument is extremely critical since this will directly have impact on the results there by affecting the conclusions drawn from the study.
  10. Results section must be explained and represented in a more explicit way and inorder to explore the factors that really contributed to the seropositivity.
  11. Statistical analyses are inadequate and in depth statistical analyses have to carried out with the data collected.
  12. The discussion of the manuscript is very very poor and authors have to completely revamp thi section and reqrite the whole discussion.

Author Response

Thank you for the opportunity to resubmit a revised copy of our manuscript entitled “SARS-CoV-2 Seroprevalence Survey in Grocery Store Workers—Minnesota, 2020-2021”. We appreciate your thoughtful comments and believe the recommended changes have strengthened the findings of this study.
The following are the changes made in the revised manuscript:

Response to changes suggested by Reviewer 1:
• The purpose of the study can be found in line 47: “The purpose of this study was to establish prevalence estimates for antibodies to SARS-CoV-2 in Minnesota grocery store workers and determine what factors (workplace, household, or county-level) may have influenced seropositivity in this target population.”
• The reasons for exploring factors that influence seropositivity are addressed in lines 51-55: “As workplaces operate in communities with varying levels of immunization, understanding the risk posed during continuous operation of grocery stores could be important to guide recommendations for managing community exposures in other settings. Additionally, the results of this study have store-level policy implications for worker safety, support, and compensation in the state of Minnesota.”
• This study was completed prior to vaccine availability, so our findings do not incorporate vaccination status.
• Our sample was a convenience sample because we did not have a comprehensive list of grocery workers available to us; this has been clarified in Line 60.
• We did not use a pre-published or validated survey for this study because the COVID-19 pandemic was a new situation that was being evaluated in real time by the scientific community. We developed this questionnaire in conjunction with our colleagues at the University of Minnesota who are experts in survey design. Given that this was an unprecedented situation, there was not a published tool available for us to use that would serve our intended purpose. This has been clarified in lines 67-71.
• We have made minor changes to the results section to make our findings more clear
• We have made organizational changes to the discussion to improve clarity

Reviewer 2 Report

According to your results, the strongest predictor of seropositivity was having a Household member with a prior positive diagnostic. This is an important factor that should be considered in statistical analysis and described in details

Author Response

Thank you for the opportunity to resubmit a revised copy of our manuscript entitled “SARS-CoV-2 Seroprevalence Survey in Grocery Store Workers—Minnesota, 2020-2021”. We appreciate your thoughtful comments and believe the recommended changes have strengthened the findings of this study.
The following are the changes made in the revised manuscript:

To further consider the impact of having a household member with a prior positive diagnostic test, we have included this predictor in our final model and have elaborated on the importance of this finding in the discussion.
• We have added this change to the methods section (Line 113-114)
• We have restructured the discussion section of this manuscript to emphasize this association

Reviewer 3 Report

The authors describe factors that had influenced seropositivity, prevalence of antibodies to SARS-Cov-2 among grocery workers in Minnesota.

The study was done very thoroughly and the results are very clearly explained. The manuscript can be improved if the conclusion of the study can be elaborated to support the argument.

Authors differentiate the population into 2 groups based on job responsibilities - direct customer contact and no direct customer contact. It will be more interesting for the readers if the authors can speculate how the interaction between the 2 groups increases the incidence of COVID-19 and spread among the grocery workers. 

Author Response

Thank you for the opportunity to resubmit a revised copy of our manuscript entitled “SARS-CoV-2 Seroprevalence Survey in Grocery Store Workers—Minnesota, 2020-2021”. We appreciate your thoughtful comments and believe the recommended changes have strengthened the findings of this study.
The following are the changes made in the revised manuscript:

  • We have elaborated upon our conclusions to heavily emphasize the broader impact of our findings
  • We state in our study limitations (Line 179-181) that given the cross-sectional design of our study, we were not able to examine patterns of disease transmission between workers.

Reviewer 4 Report

Dear authors please see below my comments.

Abstract: add how many participants (n=)

Materials and methods:  add again the number of participants, percentages of women to men, age range is missing. These details are missing at the beginning of section.

Discussion: a comparison between women and men, generally the gender impact is missing.

Author Response

Thank you for the opportunity to resubmit a revised copy of our manuscript entitled “SARS-CoV-2 Seroprevalence Survey in Grocery Store Workers—Minnesota, 2020-2021”. We appreciate your thoughtful comments and believe the recommended changes have strengthened the findings of this study.
The following are the changes made in the revised manuscript:

• We have made the abstract more clear and included the number surveyed and the number tested
• The information requested of the results section is outlined in the methods section
• We did not find any association between gender and seropositivity (Table 1). As such we have not restructured the discussion to focus on gender.

Round 2

Reviewer 1 Report

Authors have addressed reviewer's comments adequately.

Can be accepted after English language editing.

Author Response

Our manuscript has been checked by a native English-speaking colleague to carry out English language editing. From this review, we have adjusted the order of our references to reflect order of citation in the manuscript. We have also reorganized the results and discussion sections to improve readability and clarity. We re-titled Figure 2 to provide a more accurate description of the figure. Minor grammatical mistakes and punctuation errors were corrected.